# Types of Milk and/or Its Substitutes Given to Children (6–36 Months) in Nurseries in Poland: Data from the Research and Education Project “Eating Healthy, Growing Healthy”

**DOI:** 10.3390/ijerph15122789

**Published:** 2018-12-09

**Authors:** Anna Harton, Joanna Myszkowska-Ryciak

**Affiliations:** Department of Dietetics, Faculty of Human Nutrition and Consumer Sciences, Warsaw University of Life Sciences (WULS), 159C Nowoursynowska St, 02-776 Warsaw, Poland; joanna_myszkowska_ryciak@sggw.pl

**Keywords:** nurseries, infants, toddlers, milk, breast milk, milk substitutes

## Abstract

The purpose of the study was to evaluate the types of milk and/or its substitutes given to children (aged 6–36 months) in nurseries in Poland. *Methods*: The study was conducted in 211 nurseries across Poland. The supply of milk and its substitutes was checked in ten-day menus and inventory documents. In total, 211 ten-day menus and 2110 daily inventory reports were analyzed. Additionally, data were obtained by interviews with day-care center (DCC) directors and/or staff responsible for nutrition. *Results*: Compared to non-public nurseries, public ones were characterized by a higher average number of children, most often maintained their own kitchens, and charged a lower financial fee. Public DCCs also more often employed dietitians. The type of milk and its substitutes offered to children in nurseries was dependent on the age of the children and type of DCC. In a larger percentage of public DCCs infants received a milk formula, and in smaller percentage they received breast milk. This regularity also occurred in older children’s diets (13–36 months). In toddlers’ diets in public nurseries, cow’s milk was more common. The share of other milk substitutes in the nutrition of children from both age groups was negligible. *Conclusion*: The types of milk given to children in nurseries in Poland varied and depended on the age of children and the type of DCCs. It is necessary to provide education to DCC staff regarding the type of milk recommended for children under one year of age.

## 1. Introduction

Milk and dairy products are very important sources of calcium, high nutritional quality protein, and vitamins from the group B (mainly B2), as well as vitamins A and D. A properly balanced child’s diet should contain milk and dairy products [1,2]. Insufficient consumption of milk and dairy products can cause adverse health consequences, both in childhood and in adulthood [3]. In the case of growing children, milk avoidance in long-term is associated with a smaller stature, lower bone mineral mass, and increased fracture risk before puberty. In the case of adults it has been shown that there is a higher risk of osteoporosis after 50 years, especially in women after menopause. It is underlined that the consequences of age-related or postmenopausal bone loss on fracture risk depend on the level of peak bone mass achieved during childhood and adolescence and the rate of bone loss [4]. Recommendations regarding milk supply for children from different age groups are different [5,6]. The best choice for infants is breast milk [7]. According to the World Health Organization (WHO) recommendations, babies should be breastfed exclusively for the first six months [8]. After this time other foods should be introduced. Complementary foods (solids and liquids other than breast milk or infant formula) should not be introduced before 4 months but should not be delayed beyond 6 months [9]. The American Academy of Pediatrics (AAP) encourages mothers to continue breastfeeding until the baby is at least one year old, and as long after that as both mother and child are willing [10]. Breastfeeding is beneficial for baby and mother; its short- and long-term health benefits for both have been well documented [11,12]. There are numerous benefits of breastfeeding for a child, such as a reduced risk of hospitalization due to less frequent respiratory infections, lower risk of sudden infant death syndrome and otitis media, and a lower incidence of inflammatory bowel infections and necrotizing enterocolitis [13]. There is a protective effect of exclusive breastfeeding for allergic disease and a lower risk of obesity, diabetes, and celiac and childhood inflammatory bowel disease [13]. Studies show that breastfed babies have higher levels of cognitive function, as well as higher intelligence rates [14,15]. Breastfeeding is highly recommended, but sometimes breastfeeding may not be possible. There are some contraindications for both the mother and the child [16]. Infant formula is an industrially produced substitute for milk [17,18]. Infant formula attempts to mimic the nutritional composition of breast milk as closely as possible, and is based on cow’s and goat’s milk, as well as soya. Recommendations for children’s nutrition, including milk supply, change when the first year of life ends [16]. In accordance with the position of the Committee of the European Society for Pediatric Gastroenterology, Hepatology, and Nutrition (ESPGHAN), cow’s milk should not be used as the main drink before finishing 1 year of age and after 12 months of age the daily intake should not exceed 500 mL [9]. If there is no milk in the diet, it is worth including other products with a similar nutritional value. Good alternatives to traditional milk are other animal milks such as goat’s and sheep’s milk, as well as enriched vegetable products, for example soy, rice, or oat drinks. However, these products are recommended from the second year of life [16]. 

Polish studies show a low intake of milk by children aged 1–36 months [19]. Studies related to nutritional practices in Polish nurseries prove an insufficient supply of milk and related products on children’s menus [19,20]. A consequence of this situation can be calcium and vitamin D deficiency [19,21]. When planning the daily nutrition of children over the age of one year, milk and dairy products should be present in 2–3 meals per day [22,23]. A well-planned diet for infants and young children should be based on appropriate nutritional recommendations. Recommendations in this area should be followed not only by parents but also by other child caregivers. However, in current Polish law, there are no obligatory detailed guidelines on how to prepare meals for children in a nursery. Such a law exists for kindergartens [24]. The only provision concerns the need to comply with nutritional standards with regard to the content of nutrients [25]. There are some other recommendations available [26], however, they are not obligatory. According to Polish law regulations, employment of a dietician in nursery in not required. This confirms the necessity of research and assessment of the situation in this area. 

In Poland, at the end of 2015 there were 1967 nurseries; 100,000 children were covered by them [27]. In subsequent years, their numbers grew [28] and are still growing. Nutrition of young children in the first 1000 days is associated with metabolic programming and shaping eating habits. That is why the issue has a great importance, and greater knowledge about the dietary practices used in nurseries is needed. However, in Poland the research showing the organization and quality of child nutrition in day-care centers (DCCs) is limited. Only surveys conducted in individual institutions are available [29]. Such studies do not illustrate the situation regarding the whole country. An answer to this was the research and education project “Eating Healthy, Growing Healthy” (EHGH) implemented in Poland in 2015–2017 [30]. This project was dedicated to nurseries and kindergartens. As many as 2638 DCCs in total participated in the EHGH project (direct and indirect participation). The direct activities covered 1347 DCCs institutions. The majority of them were kindergartens, because the number of nurseries in Poland is still small [28]. The presented data are from the part of the project implemented in 211 nurseries, accounting for 10.8% of all existing nurseries in Poland in 2015 [27].

Nursery is a place where child spends many hours during the day and receives various meals. The infant’s (after 6 months of age) and toddler’s diet must contain the proper type of milk because it is still a basic product in their nutrition. It was interesting to check whether the nurseries offered children milk and/or its substitutes, and whether their type is suitable for age group. That is why the aim of the study was to analyze the assessment of milk and/or its substitutes served to children (6–36 months) in nurseries in Poland. 

## 2. Materials and Methods

### 2.1. General Information

The project EHGH was funded by the Danone Ecosystem Fund [30]. The purpose of the project was to improve the nutrition of children in DCCs as a result of the nutritional training of staff conducted using direct methods (by specially trained educators) and also indirectly (access to educational materials online). All activities carried out in the project were totally free of charge for DCCs; participation in the project was voluntary. Information about the project along with an invitation to participate were sent to childcare facilities across Poland (mailing lists were obtained from institutions related to care and education). In addition, information about the project was posted in dedicated information channels (magazines, websites).

No personal data were collected about children attending care facilities or institution staff. Applicants were informed about the aim and scope of the program, and the possibility of withdrawing from it at any stage without giving any reason.

Institutional Review Board approval was not necessary for this project, because it was not deemed to represent human subject research according to the University of Life Sciences Center Institutional Review Board. 

### 2.2. Study Participants

From the beginning of 2015 to the end of 2017, the EHGH project included 248 institutions taking care of small children, including public and non-public nurseries from all over Poland. The inclusion criteria for DCCs for the study were: cooperation of the center with an educator from EHGH, sharing of ten-day menus and inventory documents, staff working full-time (more than 5 h a day), and provision of full-board nutrition (defined as more than two main meals (breakfast, lunch) and one snack). Finally the inclusion criteria were fulfilled by 211 nurseries.

### 2.3. Analysis of Milk and Its Substitutes

The type of milk products served with meals in nurseries in Poland was assessed based on ten-day menus (10 consecutive days) and daily inventory documents (list of products used to prepare meals for children for each day). For this purpose, 211 ten-day menus and 2110 daily inventory reports were analyzed. Additionally, data were obtained by interviews with DCC directors and/or staff responsible for nutrition. All interviews were conducted by specially trained educators and concerned the nutrition and nutritional practices in DCCs. Interviews were performed only to verify information obtained from the menus/inventory reports (in particular the use of breast milk were examined). This analysis focused on the type of milk products served with meals or available for children between meals, such as breast milk, cow’s milk, and its substitutes (rice drink, soy drink, oat drinks etc.) as well as infant and junior formula. The study did not assess the amount of products served, but only the fact that they appear in the children’s menu and the variety of milk and its substitutes. For milk and its substitutes only qualitative data were collected. 

### 2.4. Statistical Analysis

All data were processed statistically using Statistica version 13.1 (Copyright^©^StatSoft, Inc., 1984–2014, Cracow, Poland). Table 1 presents qualitative and quantitative data such as means, standard deviations (SD), medians, minima, and maxima were calculated for the number of children attending DCCs and the cost of meals per child/day (financial fee), while for other categories only the numbers and percentages of DCCs are presented. All data were analyzed in the total group as well as for the type of DCCs (public vs. non-public). Due to the lack of a normal distribution of quantitative data (the Shapiro–Wilk statistic test for testing the normality was used), we used the Mann–Whitney U test, and in addition for qualitative data the χ-squared Pearson test was used. 

Data for the type of milk products are presented in two age groups: children under 1 years of age and children aged 13–36 months (only qualitative data). In that case statistical significances for qualitative variables were determined using the χ-squared Pearson test. For all statistics the differences were considered significant at *p* < 0.05.

## 3. Results

### 3.1. General Characteristics of DCCs

The study involved 211 nurseries from all over Poland attended by 15,253 children under the age of 3; infants accounted for only 6.5% of the total group. Table 1 presents the general characteristics of the total number of DCCs and the type of institution (public vs. non-public DCCs). Significant differences between types of DCCs were noted considering the total number of children, the number of children below one year of age, type of kitchen, and the financial fee, as well as person responsible for nutrition in the DCCs (Table 1). In the case of public DCCs, it was shown that the person responsible for planning child nutrition most often was a finance manager (109 DCCs—75%), including 73 of them with full-time, centralized supervision by a dietitian. In none of the public nurseries was there any indication of there being a dietitian especially employed for this task. In the case of non-public DCCs, dietitians were reported in nine cases (14%), but they were employed not by the nurseries but by the catering company providing food to the facility. The second most common person responsible for nutrition was the cook (seven DCCs—11%). The small number of reports of the finance manager being the responsible person (only three DCCs—5%) results from the fact that in non-public institutions such people are not employed.

### 3.2. Type of Milk Products and Substitutes Offered to Children in Nurseries Considering the Age and Type of DCCs

The type of milk products and substitutes offered to children in nurseries considering the age and type of DCCs is presented in Table 2. Considering the type of milk supplied to infants in all DCCs, it was found that most often it was infant formula followed by cow’s milk and then breast milk. Milk substitutes were given to infants very sporadically. After considering the type of the DCC, significant differences were found. It was noted that a higher percentage of public DCCs supplied children with infant formula compared to non-public DCCs, while cows’ milk and breast milk were given less frequently. Considering the type of milk products and substitutes offered in nurseries to children aged 13–36 months, it was noted that cow’s milk was served most often. In turn, the shares of infant formula and breast milk were already smaller. Similar changes in the structure of the type of milk were noted in both public and non-public DCCs. However, both types of nurseries differed significantly in terms of the type of these products. The differences are presented in Table 2.

## 4. Discussion

In Poland, children spend up to 10 h a day in nurseries [25]. That is why the institution must provide adequate food [26]. The daily diet should be based on the latest recommendations [6] and be properly balanced [1,2]. Milk and dairy products are a very important group but the type of milk should be adequate for age of children. Small amounts of cow’s milk should be used to prepare complementary foods, but it should not be used as the main milk product until the age of 12 months [9]. In the current study, many DCCs used cow’s milk in menus for infants. Serving this milk was more common in non-public nurseries. Premature introduction of cow’s milk, even to one-month-old infants, has been noted in other Polish studies [19]. According to the position of the ESPGHAN Nutrition Committee [9] as well as the AAP [31], at present there is no evidence scientifically justifying elimination or delayed introduction of potentially allergenic foods, including cow’s milk, for healthy children. However, consumption of cow’s milk by infants and toddlers has adverse effects on their iron stores [32]. It is probable because of the low iron content on cow’s milk which makes it difficult for infants to obtain the amounts of iron needed for growth. Other authors have also investigated the problem of iron deficiency in infants and toddlers [33]. That is why in nutrition for children over 6 months, when iron reserves are being depleted [34], it is more important to introduce iron-rich foods [35,36]. 

Also sheep’s and unmodified goat’s milk are not recommended for infants because of their high protein and mineral contents as well as low levels of folate. Consuming such milk increases the risk of folic acid deficiency anemia and/or B_12_ vitamins [37]. In the current study, the use of animal milks, other than cow’s milk, and other milk substitutes (plant drinks) in the nutrition of infants was sporadic. The same situation was observed in the case of toddlers’ menus. A literature review on cow’s milk and other types of milk, especially goat’s milk, proves the benefits and disadvantages of their use with children of an inappropriate age [38,39]. 

Human milk is the basis of an infant’s diet [16]. Serving breast milk in a nursery is not impossible; however, it is based on many principles, including the safety of administering breast milk [40]. Those nurseries in which there are breastfed children, in accordance with regulations, must provide the right conditions for storage of this type of milk [41]. For this reason, in everyday practice, feeding children with the use of mother’s milk in nurseries takes place relatively less frequently than with the use of milk formula. This also is confirmed by the results of current research. It has been shown that supply of breast milk was more common in non-public DCCs compared to public ones. This could be a result of the smaller groups of children. Nutrition in a small group of children can be more personalized. In this case, mothers may have a greater impact on the staff of the facilities. Breastfeeding is widely recognized as the best option for infants. However, practices within the WHO European Region are far from compliant with WHO recommendations [42] and the proportion of infants being exclusively breastfed decreases with the age of the infants. The authors proved that the prevalence of exclusive breastfeeding (EBF) under 6 months varied in different countries (median 23%); higher rates were reported in Kyrgyzstan and lower ones in Bulgaria (2%). This rate reported in Poland was only 4% [42,43]. In terms of breastfeeding at 6 months, in one study the authors noted rates between 1% and 49% (median 13%); the highest rate was seen in Slovakia (49%) and the lowest in Finland and Greece as well as the United Kingdom (only 1%) (from Poland no data were shown) [42]. Other data showed that in Poland the proportion of infants aged 7–12 months that was exclusively breastfed was 6% [19]. A lower proportion of breastfeeding in feeding older babies is a natural consequence of the change in recommendations [16]. With the age of the child, his nutritional needs changes and the mother’s milk is no longer sufficient. Between the 17th and 26th weeks of the child’s life, it is necessary to introduce complementary food [9]. 

In the case of children attending nurseries, mothers often continue to use infant formula. Mixed nutrition can also be employed—at home breastfeeding, in the nursery infant formula. Such a practice is more convenient for the mother and for the nurseries too. Giving children infant formula requires the application of multiple procedures [44,45]. In the current study, in both public and non-public nurseries, infant formula was the most frequently served type of milk for infants. Regulations in Poland until 1 January 2018 did not regulate the issue of nutrition other than the obligation to provide food in nurseries and ensure the possibility of hygienic consumption of meals by children. Thereafter, only a record of the necessity to comply with the norms [6] was introduced [46]. The above document makes no reference to breastfeeding or its promotion. The Polish Ministry of Health website [47] provides information on breastfeeding, thematic guides, and issues in breastfeeding in the perinatal care standard; promotional campaigns are carried out only in hospitals. A study in the United States in 2008 showed that only 22% of child care facilities had regulations advocating breastfeeding [48]. The authors emphasized that there is an urgent need to create new uniform legal regulations in the context of children’s nutrition in educational institutions. Another study conducted in 2016 [49] assessed state and regional variations in infant feeding regulations for early care and education and compared them to their national standards and the authors observed significant improvements. The results of other studies [50,51] argue for the need to create uniform, mandatory recommendations that will help to plan the nutrition of young children. 

In the current study, we observed changes in the assortment of milk given to children aged 13–36 of months in nurseries. The most frequently given types of milk for toddlers were cow’s milk and junior formula, the share of breast milk was the lowest. However, breast milk was given in a larger number of non-public DCCs and formula in public ones. According to recommendations, breastfeeding may be continued for at least two years or longer, as long as this is mutually desired by mother and child. In another Polish study, the authors noted that 10% of children over 1 year old were still breastfed, while more than 43% of them were received junior formula [19]. According to the European Food Safety Authority (EFSA) [52], junior milk formula is not necessary for the proper nutrition of children aged 1–3 years and has no advantage over other products contained in a normal diet that meets the basic nutritional requirements of young children. It has been shown that the share of junior formula in the nutrition of toddlers significantly impacts upon the nutritional value of their menus; for example, in terms of iron and vitamin D [53]. In a Polish study, it was noted that despite the widespread use of junior formula, diets of toddlers were still deficient in vitamin D [19]. The current study did not assess the supply of nutrients. However, such knowledge would be a valuable addition to the data on feeding practices in nurseries. These data should be supplemented in the future. 

Some differences observed in the supply of milk types between DCCs (public vs. non-public) may result from the difference noted between them such as the total number of children, the number of children below one year of age, type of kitchen, and financial fee as well as person responsible for nutrition in the DCCs. Smaller groups of children in non-public DCCs and a higher percentage of infants in total group may be associated with more frequent use of breast milk in infant nutrition. On the other hand, the greater resources allocated to children’s nutrition often results in the use of catering which is not always appropriate for each age group. In addition, the more frequent presence of a dietician in non-public DCCs often results not from the fact that dietician is employed by a nursery but by a catering company, which does not necessarily lead to better feeding of children in a nursery. Nutritionists working in catering companies deal with collective nutrition of various age groups and do not always specialize in feeding of infants. In public DCCs, the most common person of reference for nutrition is financial officer–purser, who does not need to have higher education or be a specialist in the fields of dietetics. According to Polish law regulations employment of a dietician in nursery in not required. In the absence of a dietician, nutrition is planned by different people, without specialist education in this fields, which is confirmed by other studies and publication [50,54].

Data collected in this article is unique because in recent years in Poland there have been no studies carried out on such a large scale. Earlier research concerned only individual facilities. The data clearly show that DCCs make some mistakes in infant nutrition as well as the intensive promotion of breastfeeding implemented in Poland does not translate into the supply of breast milk in nurseries. This situation proves the need to update the knowledge of the staff of the DCCs, especially in relation to infant nutrition. It is also reasonable to further disseminate knowledge about breastfeeding and translate into everyday practice. Employing dietitians in nurseries would be helpful for staff and useful for child nutrition.

### Strengths and Limitations

A major strength of this study is the large sample of nurseries from across Poland. The study includes public and non-public institutions. The majority of surveys available in Poland cover only individual institutions from selected regions of the country. The method of data collection based on a large number of documents (ten-day menus from consecutive days and daily inventory reports) acquired from nurseries is also unique. Additionally, data were obtained by interviews with DCC directors and/or staff responsible for nutrition. All interviews were conducted by trained educators based on a validated questionnaire.

Our research has some limitations. Although the study covered a large number of institutions (public and non-public) from all over the country, these are not representative of all DCCs in Poland due to the inclusion methods (DCCs enrolled to the program). The study did not assess the amount of products served, but only the fact that they appear in the children’s menu and the variety of milk and its substitutes. 

## 5. Conclusions

Infant and toddler’s diets should be well balanced, including the share of milk and its types. Recommendations for milk supply should be adjusted to the child’s age and be met at home and the nursery where the child spends many hours during the day. The current study showed that the type of milk products given to children in nurseries in Poland was varied and depended on the age of children and the type of DCCs. Some of the changes in the type of milk served to children over one year old in the examined nurseries can be considered compliant with recommendations, but the use of cow’s milk in infancy does not comply with standards. In order to avoid certain irregularities, persons responsible for nutrition in nurseries, both public and non-public, should be continually trained in child nutrition. 

## Figures and Tables

**Table 1 ijerph-15-02789-t001:** General characteristic of day-care centers (DCCs).

Category	Total DCCs (*n* = 211)	Public DCCs (*n* = 145)	Non-Public DCCs (*n* = 66)
**All children**			
Number/%	15,253/100	13,181/86	2072/14
Mean ± SD	72 ± 41.8	91 ± 34.7 *	31 ± 22.7
Median; Minimum–Maximum	94; 5–210	95; 20–210	24; 5–114
**Children under 1 years of age**			
Number/%	987/6.5	835/6.3	152/7.3
Mean ± SD	4.7 ± 8.2	5.8 ± 8.7 *	2.3 ± 6.4
Median; Minimum–Maximum	4; 0–94	2; 0–94	0; 0–40
**Type of kitchen ****			
Own kitchen: Number/%	149/71	138/95	11/16.6
Internal catering: Number/%	6/3	4/3	3/4.6
External catering: Number/%	51/24	3/2	47/71.2
Mixed (kitchen and external): Number/%	5/2	0/0	5/7.6
**Financial fee**			
Per 1 child/day/PLN ^1^			
Mean ± SD	7.0 ± 2.5	5.8 ± 1.2 *	9.7 ± 2.3
Median; Minimum–Maximum	6; 4–15	6; 4–12	10; 5–15
**Person responsible for nutrition in the DCCs ****			
Finance manager: Number/%	112/53	109/75	3/4.6
Dietician: Number/%	9/4	0/0	9/13.6
Cook: Number/%	14/7	7/5	7/10.6
Nurse: Number/%	6/3	6/4.1	0/0
No special trained person: Number/%	24/11	18/12.4	6/9.1
No person reported: Number/%	46/22	5/3.5	41/62.1

^1^ PLN (Polish zloty) = ~0.23 EUR (related only to nutrition); * significant differences between type of DCCs (Mann–Whitney U test, Number of all children *p* = 0.000, Children <1 year of age *p* = 0.000, Financial fee *p* = 0.000); ** significant differences between type of DCCs (χ-squared Pearson test, type of kitchen *p* = 0.000, person responsible for nutrition in the DCCs *p* = 0.000).

**Table 2 ijerph-15-02789-t002:** Type of milk products and substitutes offered to children in nurseries considering the age and type of day-care centers (DCCs).

Type of Milk vs. Age of Children	Total DCCs (*n* = 211) ** (Number/% of DCCs)	Public DCCs (*n* = 145) ** (Number/% of DCCs)	Non-Public DCCs (*n* = 66) ** (Number/% of DCCs)	*p*-Value
**Children under 1 years of age *****				
Breast milk	15/7.1	6/4.1	9/13.6	0.0128 *
Infant formula	145/68.7	115/79.3	30/45.5	0.0000 *
Cow’s milk	27/12.8	15/10.3	12/18.2	NS
Other milk substitutes	2/0.01	2/0.01	0/0	NS
**Children 13–36 months**				
Breast milk	11/5.2	4/2.8	7/10.6	0.0174 *
Junior formula	123/58.3	102/70.3	21/31.8	0.0000 *
Cow’s milk	182/86.3	134/92.4	48/72,7	0.0002 *
Other milk substitutes	6/0.03	2/0.01	4/0.1	NS

* significant differences public DCCs vs. non-public DCCs (χ-squared Pearson test *p* < 0.05); ** multiple replies were possible for DCCs, *** numbers do not add up to 211, because some DCCs do not provide care for infants, NS—not statistically significant (*p* > 0.05).

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
