# Peer review of "Types of Milk and/or Its Substitutes Given to Children (6–36 Months) in Nurseries in Poland: Data from the Research and Education Project “Eating Healthy, Growing Healthy”"

_ijerph, 2018, doi:10.3390/ijerph15122789_

Round 1

Reviewer 1 Report

1. In general, the paper needs to be edited by a native English speaker throughout to adhere to conventions of written English.

2. Introduction (Line 32, Line 36): It would be useful to the reader to expand on what adverse health consequences and other foods are?

3.  Introduction (Line 33): Reference # 4 needs to be updated.

4. Introduction (Line 34): Please clarify why breast milk is the best choice for infant?

5. Introduction (Line 39-42): Please revise this statement for clarity.

6.  Introduction: It was unclear based on the background and aims what gaps in the literature the study is addressing? A clearly stated rationale would strengthen the manuscript.

 7.  Introduction: What are the current nutritional guidelines for meal prepared for Polish children in childcare centers?

8.  Introduction: There is lack of previous studies to support the aim of the study.

9.   Methods: The study design is not clear to me. Was it cross-sectional or longitudinal?

10. Methods: Much information should not be presented here (Line 86-90, Line 91-94; Line 96-97). I suggest moving it to the introduction section.

11. Methods (Line 105-114). This section should be presented with major details. There is definitely not enough information on decade menus and daily inventory documents. It is not clear how the data is collected by directors/trained educators? How the data is analysed? What does questionnaire stand for?

12. Methods (Line 115-125). This section should be revised for clarity. Why data is presented in two age groups? Not sure what qualitative and quantitative variables mean? Please clarify the choice of using Chi2 and U Mann-Whitney test.

13. Results: The meaning of many sentences is unclear, please revise for clarify. P-value should be clearly presented in the text. Results of U Mann-Whitney test were not reported either in the table or text.

14. Discussion: Unfortunately, the discussion lacks cohesion. In general, this section is very week in its current format and some paragraphs are unnecessary. e.g. should the first two paragraphs present here? The authors should clarify what is this study's unique contribution to the literature on this topic. The authors don't clearly mention the implications of this study.

Author Response

Response to the Reviewers

Manuscript ID ijerph-387160

Type: Article

Title: What Type of Milk and/or its Substitutes Are Given to Children (6–36 Months) in Nurseries in Poland? Data from the Research and Education Project "Eating Healthy, Growing Healthy" Authors: Anna Harton, Joanna Myszkowska-Ryciak

Assistant Editor: Elena Martinez Batalla

Dear Mrs. Batalla,

We are very grateful for all remarks, comments and suggestions to our manuscript. Please find below the Authors’ responses to each of the Reviewer comments.

Kind regards,

Anna Harton

REVIEWER 1

1. In general, the paper needs to be edited by a native English speaker throughout to adhere to conventions of written English.

Authors’ response: Presented article was revised and checked by professional English editing service available in Poland (the certificate has been attached to the system). Additionally, by answering the reviews, we have made a few language corrections. However, following the suggestion in the case of acceptance, the text will be forwarded to the MDPI English Editing Service if necessary.

2. Introduction (Line 32, Line 36): It would be useful to the reader to expand on what adverse health consequences and other foods are?

Line 32 Authors’ response: As suggested, more information was added.

To complete this section, this section is now reads [lines 34-40]:In the case of growing children, avoidance milk in long-term is associated with smaller statue, lower bone mineral mass, and increased fracture risk before puberty. In case of adults the authors pay attention to bone health and showed higher risk of osteoporosis after 50 years, especially in women after menopause. It is underlined that the consequences of age-related or postmenopausal bone loss on fracture risk depend on the level of peak bone mass achieved during childhood and adolescents and the rate of bone loss [4]”.

Manuscript has been supplemented with an additional literature: Rizzoli, R., Bianchi, M.L., Garabedian, M., McKay, H.A., Moreno, LA. Maximizing bone mineral mass gain during growth for the prevention of fractures in the adolescents and the elderly. Bone 2010, 46, 294–305. doi: 10.1016/j.bone.2009.10.005

Line 36 Authors’ response: As suggested, more information was added.

To complete this section, this section is now reads [lines 43-45]: “After this time other foods should be introduced - complementary foods (solids and liquids other than breast milk or infant formula) should not be introduced before 4 months but should not be delayed beyond 6 months [9]”

The new quotation has been added to the completed content instead of position number 17 and 35.

Fewtrell, M., Bronsky, J., Campoy, C., Domello¨f, M., Embleton, N., Fidler Mis, N., Hojsak, I., Hulst, J.M., Indrio, F., Lapillonne, A., Molgaard, Ch. Complementary Feeding: A Position Paper by the European Society for Paediatric Gastroenterology, Hepatology, and Nutrition (ESPGHAN) Committee on Nutrition. J. Pediatr. Gastroenterol. Nutr. 2017, 64, 119–132

3. Introduction (Line 33): Reference # 4 needs to be updated.

Authors’ response: It has been done.

Now we have: position 5: Optimizing Bone Health and Calcium Intakes of Infants, Children, and Adolescents. Pediatrics 2006, 117(2). Available online: http://pediatrics.aappublications.org/content/117/2/578.long (accessed on November 27 2018).

4. Introduction (Line 34): Please clarify why breast milk is the best choice for infant?

Authors’ response: The explanation of this is below lines: 47-54

5. Introduction (Line 39-42): Please revise this statement for clarity

Authors’ response: This record was unfortunate, now it has been corrected on “Children in the first year ….” [line 48]

6. Introduction: It was unclear based on the background and aims what gaps in the literature the study is addressing? A clearly stated rationale would strengthen the manuscript.

Authors’ response: Data on children feeding practises in homes or in nurseries, individual studies that exist concern a small number of DCCs. In recent years, more and more DCCs in Poland are being established and more children are undergoing nursery care. There are no precise and detailed regulations, including the obligation to employ dietician.

To clarify this it now reads:

[line: 67] Polish studies show a low intake of milk by children aged 1–36 months [19].

[lines: 85] Only surveys conducted in individual institutions are available [29].

[lines: 73-79] However, in the Polish current law, there are no obligatory detailed guidelines on how to prepare meals for children in a nursery. Such law exists for kindergartens [24]. The only provision concerns the need to comply with nutritional standards with regard to the content of nutrients [25]. There are some other recommendations available [26], however, they are not obligatory. According to Polish law regulations employment of a dietician in nursery in not required. This confirms the necessity of research and assessment of situation in this area.

[lines: 80-81]. In Poland, at the end of 2015 there were 1,967 nurseries; 100,000 children were covered by them [27]. In subsequent years, their number grew [28] and it is still growing.

7. Introduction: What are the current nutritional guidelines for meal prepared for Polish children in childcare centers?

Authors’ response: [lines: 73-77]: “However, in the Polish current law, there are no obligatory detailed guidelines on how to prepare meals for children in a nursery. Such law exists for kindergartens [24]. The only provision concerns the need to comply with nutritional standards with regard to the content of nutrients [25]. There are some other recommendations available [26], however, they are not obligatory.”

8. Introduction: There is lack of previous studies to support the aim of the study.

Authors’ response: Data on children feeding practises in homes or in nurseries, individual studies that exist concern a small number of DCCs.

To clarify this it now reads:

[line: 67] Polish studies show a low intake of milk by children aged 1–36 months [19].

[line: 85] Only surveys conducted in individual institutions are available [29].

9. Methods: The study design is not clear to me. Was it cross-sectional or longitudinal?

Authors’ response: The project had longitudinal character, however the presented part was cross-sectional type.

10. Methods: Much information should not be presented here (Line 86-90, Line 91-94; Line 96-97). I suggest moving it to the introduction section.

Authors’ response: It has been done following the suggestion.

11. Methods (Line 105-114). This section should be presented with major details. There is definitely not enough information on decade menus and daily inventory documents. It is not clear how the data is collected by directors/trained educators? How the data is analysed? What does questionnaire stand for?

Authors’ response: It has been explained in lines: 101-108, 116-121, 123-134

12. Methods (Line 115-125). This section should be revised for clarity. Why data is presented in two age groups? Not sure what qualitative and quantitative variables mean? Please clarify the choice of using Chi2 and U Mann-Whitney test.

Authors’ response: It has been explained in lines: 136-147

13. Results: The meaning of many sentences is unclear, please revise for clarify. P-value should be clearly presented in the text. Results of U Mann-Whitney test were not reported either in the table or text.

Authors’ response: Statistical differences are described above the tables and marked in individual tables and described under them.

For Table 1 – [lines 153-155]: “Significant differences between types of DCCs were noted considering the total number of children, the number of children below 1 year of age, type of kitchen, financial fee as well as person responsible for nutrition in the DCCs (Table 1)”

Results of U Mann-Whitney have been marked with * and along with explanation are placed below the Table 1.

[lines: 166-167]: “* significant differences between type of DCCs (U Mann-Whitney test, Number of all children p = 0.000, Children <1 year of age p = 0.000, Financial fee p = 0.000)”

Results of Chi 2 Pearson test have been marked with ** and are placed below Table 1

[lines:168-169]; “** significant differences between type of DCCs (Chi 2 Pearson test, Type of kitchen p=0.000, Person responsible for nutrition in the DCCs p = 0.000).”

In Table 2 there are only results for qualitative data where only Chi 2 Pearson test is used - significant differences are described above the table: lines: 171-181

[lines: 184-185]: „* significant differences public DCCs vs. non-public DCCs (chi 2 Pearson test p < 0.05)”

Additionally below table 2 some more explanation for NS have been added

[line: 185]: „NS – not statistically significant (p>0,05)”

14. Discussion: Unfortunately, the discussion lacks cohesion. In general, this section is very week in its current format and some paragraphs are unnecessary. e.g. should the first two paragraphs present here? The authors should clarify what is this study's unique contribution to the literature on this topic. The authors don't clearly mention the implications of this study.

Authors’ response: According to the suggestions, the first paragraphs have been removed, the layout has been changed and the content has been ordered. The whole manuscript was supplemented with information on study's unique contribution to the literature and the implications of this study has been clarified.

Thank you very much for your time and all the valuable comments.

Reviewer 2 Report

Comments to the Authors

This manuscript addresses an important issue in childhood nutrition in nurseries.

Comments:

The survey on the breast milk and its substitutes was carried out using EHGH project data. Readers will want to know the collection method of 248 nurseries for the EHGH project. Because author wrote that these are not representative of all DCCs in Poland in limitation section.

One of informative results was that 27 nurseries gave infant cow’s milk. Appropriate interpretation of the result is necessary to develop discussion. Sound and creative conclusion will be derived from result-based discussion.

Although Institutional Review Board Approval may not be necessary, the permission of data using from EHGH project organization will be necessary.

There were several passages that were not clear for me:

1) Page 1 line 39-42, “Infants in the first years …..” .

    The content of the passage is correct when breastfed infant is compared to formula-fed infant.

2) How different following inclusion criteria?

     Page 2 line 90; the inclusion criteria

     Page 3 line 101; Additional including criteria

Page 5  Table 2

The total number of nurseries with breast milk (15), infant formula (145), cow’s milk (27) and other milk substitute (2) is 189. Were milk and its substitute given to infants or not in another 22 nurseries ?

Author Response

Response to the Reviewers

Manuscript ID ijerph-387160

Type: Article

Title: What Type of Milk and/or its Substitutes Are Given to Children (6–36 Months) in Nurseries in Poland? Data from the Research and Education Project "Eating Healthy, Growing Healthy" Authors: Anna Harton, Joanna Myszkowska-Ryciak

Assistant Editor: Elena Martinez Batalla

Dear Mrs. Batalla,

We are very grateful for all remarks, comments and suggestions to our manuscript. Please find below the Authors’ responses to each of the Reviewer comments.

Kind regards,

Anna Harton

REVIEWER 2

Review Report Form:

English language and style (x) Extensive editing of English language and style required 

Authors’ response: Presented article was revised and checked by professional English editing service available in Poland (the certificate has been attached to the system). Additionally, by answering the reviews, we have made a few language corrections. However, following the suggestion in the case of acceptance, the text will be forwarded to the MDPI English Editing Service if necessary.

Comments:

The survey on the breast milk and its substitutes was carried out using EHGH project data. Readers will want to know the collection method of 248 nurseries for the EHGH project. Because author wrote that these are not representative of all DCCs in Poland in limitation section.

Authors’ response: We agree with the comment and add some more information about DCCs enrolment to the project. It now reads as [page 3, line 105]: Information about the project along with an invitation to participate were sent to childcare facilities across Poland (mailing lists were obtained from institutions related to care and education), in addition, information about the project was posted in dedicated information channels (magazines, websites).

The surveyed group is not representative of all DCCs in Poland because the data was collected from among nurseries participated in the EHGH project. The initial number of the facilities enrolled to the project was 248 but then due to non-fulfillment of the inclusion criteria, the final number was 211.

To clarify the situation, this section is now reads [page 3, lines 116-121]:

From the beginning of 2015 to the end of 2017, the EHGH project included 248 institutions taking care of small children, including public and non-public nurseries from all over Poland. The inclusion criteria for DCCs for the study were: cooperation with an educator from EHGH, sharing decade menus and inventory documents as well as working on full-time (more than 5 hours a day) with full-board nutrition, was defined as more than 2 main meals (breakfast, lunch) and 1 snack. Finally the inclusion criteria were fulfilled by 211 nurseries.

One of informative results was that 27 nurseries gave infant cow’s milk. Appropriate interpretation of the result is necessary to develop discussion. Sound and creative conclusion will be derived from result-based discussion.

Authors’ response: Following this attention, we have modified the discussion to emphasize the aspect of giving cow's milk to young children.

Although Institutional Review Board Approval may not be necessary, the permission of data using from EHGH project organization will be necessary.

Authors’ response: The research part of the project was planned and implemented only by WULS employees (authors of manuscript). The data obtained as part of the project in accordance with the project contract are exclusive property of WULS. The data can be published is information about the project and the grant provider are attached.

There were several passages that were not clear for me:

1) Page 1 line 39-42, “Infants in the first years …..” .

    The content of the passage is correct when breastfed infant is compared to formula-fed infant.

Authors’ response: This record was unfortunate, now it has been corrected on “Children in the first year ….” [page 1, line 48]

2) How different following inclusion criteria?

     Page 2 line 90; the inclusion criteria

Authors’ response: To clarify the situation, this section is now reads [page 3, lines: 117-120]: The inclusion criteria for DCCs for the study were: cooperation with an educator from EHGH, sharing decade menus and inventory documents as well as working on full-time (more than 5 hours a day) with full-board nutrition, was defined as more than 2 main meals (breakfast, lunch) and 1 snack.

     Page 3 line 101; Additional including criteria

Authors’ response: To clarify the situation the inclusion and additional criteria were combined, additionally there were shortened as well as the text between them was removed. The whole reads like above – lines 117-120.

Page 5  Table 2

The total number of nurseries with breast milk (15), infant formula (145), cow’s milk (27) and other milk substitute (2) is 189. Were milk and its substitute given to infants or not in another 22 nurseries ?

Authors’ response:  Each nursery could provide several variants of answers, so the values do not have to add up to the total number of DCCs. We agree with the fact that it is unclear. We added information (under the table) that multiple replies were possible for DCCs. Values do not add up to a given number of DCCs, the percentage was counted as the number of answers vs the number of DCCs in individual group (total, public, non-public).

Thank you very much for your time and all the valuable comments.

Round 2

Reviewer 1 Report

No further comments. The manuscript comes up much stronger now.

Author Response

Dear Reviewer,Thank you very much for the constructive review, in fact the manuscript is much better in this version.

Kind regards, 

Anna Harton

Reviewer 2 Report

Although the manuscript has been improved in the revised version,

I still need your explanation on a passage and Table2.

Page 1 line 39-42 in the previous version;   

Page 1  line 150-153 in the revised version

Children in the…….   Please confirm the content is correct or unfortunate.

Table 2

Regarding on children under aged one year,

Total number of nursery is 211

Breast milk, infant formula, cow’s milk, other milk substitute were provided in 189 (multiple replies) out of 211 nurseries

Please confirm what happen to other 22 nurseries.

Author Response

Response to the Reviewers

Manuscript ID ijerph-387160

Type: Article

Title: What Type of Milk and/or its Substitutes Are Given to Children (6–36 Months) in Nurseries in Poland? Data from the Research and Education Project "Eating Healthy, Growing Healthy" Authors: Anna Harton, Joanna Myszkowska-Ryciak

Assistant Editor: Elena Martinez Batalla

Dear Mrs. Batalla,

We are very grateful for all remarks, comments and suggestions to our manuscript. Please find below the Authors’ responses to each of the Reviewer comments.

Kind regards,

Anna Harton

REVIEWER 2

Comments:

Although the manuscript has been improved in the revised version,

I still need your explanation on a passage and Table2.

Page 1 line 39-42 in the previous version;   

Page 1  line 150-153 in the revised version

Children in the…….   Please confirm the content is correct or unfortunate.

Authors’ response: In fact, the previous explanation was not fortunate. The sentence uses a certain mental shortcut referring to the short-term benefits of breastfeeding concerning infants, and long-term benefits concerning children above one year of age.

To make it more readable, this section is now reads:

[lines: 48-51]: “There are numerous benefits of breastfeeding for a child, such as a reduced risk of hospitalization due to less frequent respiratory infections, sudden infant death syndrome, otitis media, as well as the incidence of inflammatory bowel infections and necrotizing enterocolitis”.

Table 2

Regarding on children under aged one year,

Total number of nursery is 211

Breast milk, infant formula, cow’s milk, other milk substitute were provided in 189 (multiple replies) out of 211 nurseries

Please confirm what happen to other 22 nurseries.

Authors’ response: The data presented in Table 2 refers to the nutrition of infants and separately young children (over 1 year of age). However, in some DCCs, there were no infants (as indicated in Table 1), hence the missing data, which do not add up to 211.

We agree with the fact that it is unclear. In the table 2 a new designation has been added “***” and we added more information/description under the table in line 185:

This section is now reads: “***numbers do not add up to 211, because some DCCs do not provide care for infants”

Thank you very much for your time and all the valuable comments.